# The Impact of Corporate Innovation on Environmental Performance: The Moderating Effect of Financing Constraints and Government Subsidies

Hui Deng [1], Chuang Li [1,2,*] and Liping Wang [3]

1   Research Center for Energy Economics, Henan Polytechnic University, Jiaozuo 454000, China
2   School of Business Administration, Jimei University, Xiamen 361021, China
3   Finance and Economics College, Jimei University, Xiamen 361021, China
*   Correspondence: lichuanghpuedu@126.com; Tel.: +86-173-6592-5651

**Abstract:** As an essential means of reducing environmental stress, corporate innovation faces financial pressure and financial risk; so, whether corporate innovation contributes to environmental performance is related to the firm's external capital environment (financing constraints and government subsidies). This study explores the relationships between corporate innovation, environmental performance, and the external capital environment using 1127 observations of China's energy-intensive public companies from 2012 to 2020. The outcomes indicate that corporate innovation has a significant positive impact on environmental performance. Financing constraints provide a negative moderation of the relationship between corporate innovation and environmental performance, while government subsidies provide a positive moderation. That is, firms with high financing constraints are less likely to increase their environmental performance by innovation, while firms with high government subsidies enhance the positive influence of corporate innovation on environmental performance. The moderating effect of financing constraints varies at different levels of government subsidies, i.e., high levels of government subsidies diminish the negative moderating effect of the financing constraints between corporate innovation and environmental performance. This study's conclusions provide a reference for the government in formulating corporate innovation subsidies and financial policies and a basis for the decision-making behaviors of enterprises regarding environmental protection and economic development.

**Keywords:** corporate innovation; environmental performance; external capital environment; financing constraints; government subsidies

## 1. Introduction

The problem of environmental contamination brought about by drastic economic growth has not been effectively controlled and tackled, [1] and the rapid transformation of the corporate economy is an attractive way for China to realize high-quality and eco-friendly development. In recent years, China has placed greater emphasis on ecological protection with the enactment of the Environmental Protection Tax Law of the People's Republic of China in 2018 and the Law of the People's Republic of China on the Prevention and Control of Environmental Pollution Caused by Solid Wastes in 2020. The 14th Five-Year Plan of China sets strategic goals to achieve "peak carbon emissions" and "carbon neutrality" and specifies the main role of innovation in China's economic development. As major consumers of energy and creators of environmental problems [2], enterprises are obliged to improve their environmental performances to comply with government requirements on the environment. Innovation is a valid way for companies to realize accelerated economic transformation and high-quality development, but their innovation activities have operation management risks, financing risks, and output unpredictability risks. Innovative projects such as energy conservation and the circular economy require the

support of adequate financial resources; so, financial resource allocations, such as financing and government subsidies, are primary motivators of the enterprises' involvement in environmental activities [3,4]. Therefore, improving the external capital environment of enterprises to minimize pollution and carbon footprints has become a focus for the integration of business-driven innovation and a low-carbon economy.

Corporate innovation improves productivity and transforms patterns of economic growth; so, innovation is considered an essential driver for achieving sustainable development [5–7]. For example, Sohag et al. [8] found that corporate innovation mitigates environmental pollution through cleaner production, environmental management, and full-cost accounting. However, because undesirable output varies in proportion to desirable output [9], innovation scales up economic expansion and consumes large amounts of resources, and thus, innovation is also one of the major causes of large pollutant emissions [10]. In summary, it is clear that with regard to whether corporate innovation has improved, environmental performance is an issue that deserves deeper consideration.

As there is a lack of consensus on the effect of corporate innovation on environmental performance, various moderating and mediating variables are considered in terms of the relevant variables affecting the focal relationship. The relevant literature has focused on environmental and organizational factors. In terms of organizational factors, scholars have used perspectives such as technological dynamism, firm size, corporate image, and organizational learning to conduct their analyses. Yu et al. [11] argued that the positive effects of exploration and exploitation on environmental behaviors are moderated by technological dynamism and firm size, respectively. Ma et al. [12] concluded that corporate image has a long-term mediated role in green-process innovation based on short- and long-term benefits. Tu and Wu [13] revealed the mediation effects of organizational learning in the process of green innovation affecting firms' competitive strengths. Regarding environmental factors, environmental regulation, group affiliation, and environmental dynamics all influence the focal relationship. Tian et al. [14] studied Chinese listed companies in environment-related industries and found that the enforcement of environmental policies contributed to enhance the positive influence of business innovation on environmental investments. Woo et al. [15] revealed a positive moderating effect of corporate group affiliation and listed the complementary assets of the focal relationship. Chan et al. [16] argued that environmental dynamics moderate the relationship between green product innovation and cost efficiency, with a stronger positive relationship between green product innovation and cost efficiency in highly dynamic circumstances. However, in terms of the amount of resources allocated, each company has budgetary constraints [17], and corporate innovations with high risk and long lead times require high investment budgets; therefore, whether corporate innovation helps to reduce environmental stress is also significantly related to the firm's external capital environment.

Financing constraints and government subsidies are important external capital environments for firms. When operators face financing constraints, innovation budgets are constrained, which in turn can affect corporate environmental practices. In addition, market failure theory and signaling theory suggest that the government should subsidize the firms' innovation activities, mitigating the shortage of innovation funds [18]. In contrast, principal–agent theory argues that innovative activity is severely hampered by the irrational allocation of government subsidy resources and the crowding out of subsidy funds [19]. Thus, whether innovation promotes corporate environmental responsibility is also influenced by government subsidies. In terms of the external capital environment, there are few studies in the literature on the moderating role of financing constraints and government subsidies; therefore, it is necessary to analyze the impact of financing constraints and government subsidies on the relationship between corporate innovation and environmental performance.

Existing studies have diverging views on whether corporate innovation improves environmental performance, and there is a gap in the examination of the relationship between corporate innovation and the environment from an external capital environment

perspective. Fewer studies have used emerging economies with inadequate financial and fiscal policy systems as a sample context. Will corporate innovation based on multiple theories of resource processes, such as resource acquisition, resource allocation and integration, and resource profitability, improve environmental performance? What are the implications of financing constraints and government subsidies on the relationship between corporate innovation and environmental performance in the context of the external capital environment? What are the changes in the influences of financing constraints with differences in the degree of government subsidies? These questions need to be thoroughly investigated. Energy-intensive enterprises are important market carriers in China and important contributors to pollution emissions; so, this paper chooses energy-intensive public companies as the subjects of study. To address the above three questions, first the relationship between corporate innovation and environmental performance is investigated, utilizing both content and empirical analyses. Second, from the external capital environment perspective, financing constraints and government subsidies are tested separately for their moderating effects on the relationship between corporate innovation and environmental performance. Finally, a further study tests the co-moderating impacts of financing constraints and government subsidies on this relationship. This study aims to promote enterprises' active environmental responsibility and to offer theoretical contributions and practical guidelines for designing policies related to solving environmental problems.

This study has four main contributions: First, the influences of corporate innovation on environmental performance and the process factors, such as resource utilization, pollution output, and the product management of innovation, are mainly studied, but the findings differ due to the different theories and perspectives applied. This study incorporates reputation theory, resource-based theory, and sustainability theory into the theoretical analysis framework, which enriches the theoretical mechanism of the influences of corporate innovation on environmental performance. Second, scholars currently focus on non-financial factors such as corporate organizational factors and the policy environment to influence the relationship between corporate innovation and environmental performance, but studies on financial factors have not been addressed. This study investigates the impact of the enterprises' external capital environment in two dimensions: financing constraints and government subsidies; it contributes to the unveiling of the policymaking mechanisms of enterprise innovation. Third, the corporate financing constraints vary with government subsidies; thus, the compound adjustment of the external capital environment affects the environmental benefits generated by corporate innovation decisions. This study further considers whether corporate financing constraints are affected by the degree of government subsidies, which is useful for exploring the structural black box of the firm's external capital environment. Fourth, most studies on the influence of corporate innovation on environmental performance are focused on developed countries with strong financial systems and political mechanisms [20,21], while fewer studies have focused on emerging economies with immature credit markets and weak policy implementation. With a unique political and economic context, China is facing progressively more severe environmental issues. Understanding the innovation and environmental behaviors of Chinese enterprises provides new ideas for government adjustments to subsidies and financial policies, and China's measures to improve corporate environmental performance provide a significant reference for developing a low-carbon economy in emerging economies.

This study is structured as listed below: the second part describes the theories and hypotheses; the third part presents the research design; the fourth part contains the empirical analysis; the fifth part consists of the research discussion; and the sixth part shows the conclusion and the implications.

## 2. Theories and Hypotheses

### 2.1. Corporate Innovation and Its Impact on Environmental Performance

Corporate innovation is based on resources, and the changing activities of resources in the innovation process have influences on environmental performance. The research

analyzes the impacts of corporate innovation on environmental performance by the resource processes of corporate resource acquisition, resource allocation and integration, and resource profitability via reputation theory, resource-based theory, and sustainability theory, respectively.

Reputation theory suggests that corporate reputation manifests itself as a unique asset in the marketplace and that corporations with high innovation capabilities have higher reputations. Studies have shown that corporate innovation can influence environmental performance through the reputation mechanism. First, to preserve reputation, enterprises are motivated to improve the effectiveness of innovation to satisfy government requirements for the environment. Thus, Gangi et al. [22] found that to uphold reputation companies are motivated to engage in innovation and environmental practices that positively affect financial and environmental indicators. Second, to preserve reputation and foster high customer loyalty, enterprises are more motivated to innovate and produce green products. As Sridhar and Mehta [23] revealed, building reputation is a key factor for firms to proactively innovate in producing environmentally friendly products, enabling increased customer cross-purchase intentions. Furthermore, an already acquired good reputation enables enterprises to establish more stable political connections and gain access to innovation resources, green development opportunities, and support. Qiu et al. [24] showed that a good reputation reduces capital costs and monitoring costs so that enterprises are more likely to attract key resources for innovation (innovation subsidies, policy incentives, etc.) and to penetrate environmentally sensitive markets.

Resource-based theory suggests that polluting emissions indicate that resources are not being used efficiently, resulting in economic waste [25]. Innovation is an important vehicle for the effective distribution of resources and is the key to transforming a short-term competitive advantage into a sustainable competitive advantage [26]. In other words, resource-based theory considers that a firm's innovation capability represents its total factor productivity, embodied in the reduction in pollution or damage to the environment from production and operation activities. Abbas and Sagsan [27] suggested that corporate innovation has shifted the economic development model from relying on production factors to being innovation-driven, reducing the pollution emissions of industrialization. Hart and Dowell [25] revealed that corporate innovation can not only reduce production costs but can also contribute to environmental improvements by reducing undesirable outputs such as wastewater, exhaust gases, and solid waste. Miao et al. [28] identified that green innovation (the particular kind of corporate innovation) allows for clean production, source pollution control, and waste recycling. Therefore, corporate innovation facilitates the optimal combination of the enterprises' production factors, reducing resource consumption and pollution emissions per unit of output.

According to sustainability theory, the core objective of environmental sustainability is to achieve economic development without burdening the environment. Scholars generally agree that environmental sustainability must be achieved through innovation. The early stage of sustainable development theory considered innovation to be the reducing of the environmental impact or the improvement of environmental performance by improving existing products or developing new products, emphasizing the balance between economic growth and natural limits. For instance, Lampikoski et al. [29] suggested that innovation focused on shrinking the environmental footprint to enhance competitiveness while preserving the environment to save valuable resources for future generations. The current stage of sustainable development theory advocates that innovation incorporates social and economic factors in improving the environment to achieve harmony between production and consumption, emphasizing the maximization of the combined benefits of the economy, society, and ecology. For example, Silvestre and Ţîrcă [6] argued that innovation can transform organizations, supply chains, communities, and the environment in a sustainable direction. Therefore, the following research hypothesis is raised:

**Hypothesis 1 (H1).** *Corporate innovation has a positive impact on environmental performance.*

## 2.2. Moderating Effect of Financing Constraints

The processes for corporate innovation to affect environmental performance are somewhat dependent on the firm's financing activities [30]. Resource-based theory suggests that adequate financial resources are critical aspects of corporate innovation. When an enterprise's innovation period is lengthy and high-risk, the reverse selection risk and moral hazard due to financing information asymmetry increase [31], making it difficult for the enterprise to acquire sufficient and sustainable financial support.

Financing capital is the pool of capital for the growth potential of innovation-focused companies [31]. With weaker financing constraints, the financial risk of enterprises can be steadily secured, which to a certain extent bridges the "gap" between the supply and demand of corporate innovation financing and achieves deeper corporate innovation [32]. Corporate innovation for environmental improvement, such as pollution prevention, energy conservation, green product design, and waste recycling, is also stimulated [33]. Hence, with low financing constraints, enterprises are able to effectively counter various environmental issues, reduce the risk of impaired reputation, and improve environmental performance.

Conversely, with stronger financing constraints, some technologies of little economic value will not be commercialized, and the opportunity cost of corporate innovation investment will be higher [32]. Therefore, with a high level of financing constraints, enterprises are less willing to commit to environmental programs with low short-term returns and high risk, resulting in lower environmental performance. Cecere et al. [34] argued that firms may forgo the long-term benefits of corporate innovation with insufficient economic conditions and affordability. Kim [35] indicated that in times of financing constraints, investors may have to abandon or delay corporate innovation activities for fear of bankruptcy, resulting in negative impacts on environmental performance. Thus, this research puts forward the following hypothesis.

**Hypothesis 2 (H2).** *Financing constraints negatively moderate the relationship between corporate innovation and environmental performance.*

## 2.3. Moderating Effect of Government Subsidies

Government subsidies are policy tools explicitly aimed at helping enterprises implement socially beneficial corporate innovation activities [36]. Energy-intensive industries, being an essential component of China's economy, enjoy substantial government subsidies, with the share of innovation subsidies in each industry being above 10% [18]. Government subsidies can influence environmental performance by supplementing the firms' innovation resources [37]. On the one hand, government subsidies have the resource property to directly provide financial support for corporate innovation, thus motivating companies to produce low-carbon products and to reduce negative environmental consequences. For example, Li et al. [38] found that government subsidies reduced corporate innovation costs and stimulated the development of low-carbon technologies, thereby mitigating carbon emissions.

On the other hand, government subsidies have signaling properties, facilitating firms in obtaining resources and support from stakeholders. According to signaling theory, government subsidies give active signals to the outside capitalists, making them receptive to increasing the share of innovation investments favorable to the environment [37,39]. For example, Lai et al. [40] showed that government subsidies not only indicate the government's appreciation for firms' innovation achievements, thus making firms more likely to receive financial support from investors for investment, they also enhance the green image of companies, making them more inclined to produce green products and services through innovation. Therefore, government subsidies reduce the financial pressure and financial risk in the corporate innovation process, facilitating companies in assuming environmental responsibility actively [41] and in turn improving environmental performance. Consequently, the following research hypothesis is offered in this study.

**Hypothesis 3 (H3).** *Government subsidies positively moderate the relationship between corporate innovation and environmental performance.*

Figure 1 shows the composition of the main concepts and research hypotheses of this study, demonstrating how corporate innovation affects environmental performance in the perspective of the external capital environment.

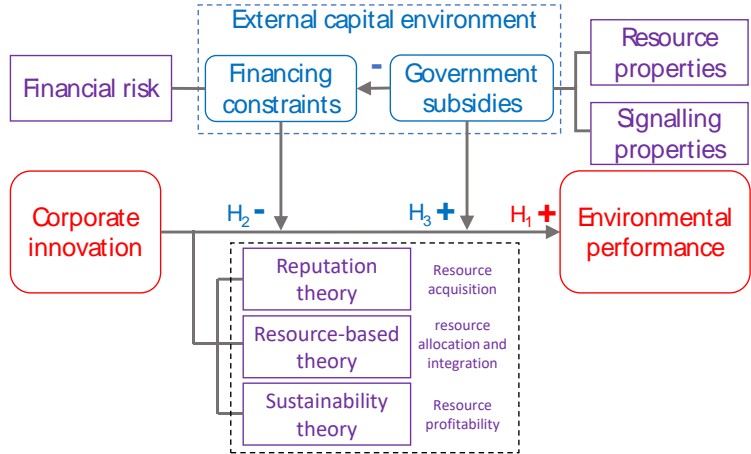

**Figure 1.** Conceptual model and research hypotheses. The plus sign "+" means a positive effect; the minus sign "-" means a negative effect; the arrow "→" connects the beginning and end of the effect, where the arrow points to the end; the straight line "-" connects the theoretical process or variable properties.

## 3. Research Design

### 3.1. Samples and Data Sources

Energy-intensive industries are the mainstay of China's economic and social development as well as significant pollution emission sources. According to the energy consumption industry ranking in China's Guide to Energy Conservation, Carbon Reduction and Upgrading in Key Areas of High Energy Consumption Industries, enterprises in the top 10 high energy consumption industries listed in the Shanghai and Shenzhen A-shares were selected to be the original objects with data from 2012 to 2020.

First, to guarantee the integrity and credibility of the research samples, the samples marked as "ST" and "PT" within the research period were excluded. Second, the listed companies with severely missing data on the study variables (except government subsidies and environmental performance) were excluded. Furthermore, the above data were supplemented by government subsidy and environmental performance data derived from content analysis. Due to the accessibility of the data, this step removed years with incomplete data on government subsidies and environmental performance. Finally, individual missing data were filled by interpolation, and continuous variables were winsorized to dispose of outliers. The final 1127 observations for 463 sample firms were obtained.

The data sources were as follows. The environmental performance data were comprehensive scores of the environmental performance indicator system (Table 1). The quantitative scores of each indicator were obtained by transforming the qualitative descriptions in the corporate annual reports, environmental reports, and social responsibility reports. The government subsidies data were aggregated from the non-operating income section in the corporate annual reports, including mainly "subsidies for scientific research", "subsidies for energy saving and emission reduction", "subsidies for technical improvement", and other items related to environmental protection. Corporate innovation data came from the Wind database. The data for the remaining variables in this study, such as financing constraints, total assets, and operating income, were obtained from the China Stock Market and Accounting Research (CSMAR) database.

**Table 1.** Environmental performance index system.

| Criterion Layer | Factor Layer | Variable Assignment |
|---|---|---|
| Environmental governance | Environmental liability report or environmental report | 0~2 |
| | "Three wastes" pollutant emissions | 0~2 |
| | Environmentally friendly charity activities | 0~2 |
| Environmental rewards and punishments | Environmental punishments | 0~2 |
| | Environmental rewards | 0~2 |
| Environmental objectives | Environmental honors | 0~2 |
| | Environmental management system certification | 0~2 |

The quantitative scoring process of each indicator: factor layer indicator content that does not exist is rated as 0; factor layer indicator content that exists but has no quantitative information is rated as 1; factor layer indicator content that exists and has quantitative information is rated as 2.

*3.2. Variable Measurement*

3.2.1. Dependent Variable: Environmental Performance

Most studies used the TRI (Toxic Release Inventory), CEP (Council on Economic Priorities), or KLD (Kinder, Lydenberg, and Domini) databases to measure environmental performance, but they provide data only for developed countries. There is no database to measure corporate environmental performance in China [42]. Therefore, some scholars adopt quantitative and qualitative approaches to measure environmental performance. Quantitative indicators mainly adopt pollution emission costs [43], environmental protection capital expenditure [44], etc., and qualitative indicators mainly adopt types of environmental rewards and penalties [45], such as whether an enterprise received environmental honors [42], etc. However, these metrics may have the following drawbacks: first, the standards of relevant costs and expenses disclosed within the company vary significantly, and quantitative indicators may suffer from conflicting definitions; second, the quantitative indicators have difficulty fully capturing the environmental management situation of the enterprise, and there are problems such as a lack of objective criteria for assigning values. Therefore, this paper reflects enterprises' environmental protection in different aspects, such as environmental policies, environmental behaviors, and environmental visions, based on the sustainability reporting guidelines (G4) published by the Global Reporting Initiative. Moreover, combined with the indicator setting of environmental performance by Henri and Journeault [46], environmental performance is divided into environmental governance, environmental rewards and punishments, and environmental objectives. The specific index selection and variable assignments are shown in Table 1. This study combines the content analysis method of Clarkson et al. [47], converts qualitative descriptions into quantitative indicators, and scores the textual content quantitatively on a scale of 0 to 2 according to the principles of undisclosed/disclosed and combined qualitative/quantitative scores, thus obtaining a composite score for each company's environmental performance.

3.2.2. Explanatory Variable: Corporate Innovation

The studies have inconsistently measured corporate innovation [48], which is mainly divided into innovation expenditures and innovation results [2]. Data such as innovation expenditures were more available and statistically consistent than innovation outcomes. Thus, innovation expenditure was used to gauge corporate innovation. To further obviate the impact of different industry sizes, the innovation expenses were logarithmized.

### 3.2.3. Moderating Variables: Financing Constraints and Government Subsidies

Hadlock and Pierce [49] measured the financing constraints of public companies using the SA index. As the SA index is minus, this paper took its absolute value as the financing constraints, with a larger absolute value representing stronger financing constraints.

The government subsidies data were aggregated from the subsidies data in the non-operating income section of the listed companies' annual reports, mainly including "subsidies for scientific research", "subsidies for energy saving and emission reduction", "subsidies for technical improvement", and other items related to environmental protection. Government subsidies were described by the logarithm of the government subsidies within the current period.

### 3.2.4. Control Variables

Several variables likely to affect changes in environmental activity were controlled for, primarily leverage (*Lev*), ownership concentration (*Cent*), growth capacity (*Growth*), profitability (*ROA*), firm size, and firm age (*Age*). Table 2 aggregates the definitions and descriptions of all variables.

**Table 2.** Variable definition.

| Attributes | Variable | Symbol | Definition |
|---|---|---|---|
| Dependent variable | Environmental performance | *EP* | Synthesis evaluation index |
| Explanatory variable | Corporate innovation | *RD* | The logarithm of innovation expenditures |
| Moderating variables | Financing constraints | *SAA* | Absolute value of SA index |
| | Government subsidies | *Govsubsidy* | The logarithm of the government subsidies |
| Control variables | Profitability | *ROA* | The rate of return on assets |
| | Ownership concentration | *Cent* | The shareholding percentage of the largest shareholder |
| | Growth capacity | *Growth* | The growth rate of operating income |
| | Leverage | *Lev* | Percentage of total debt to total assets |
| | Firm size | *Size* | The logarithm of total assets at the end of the year |
| | Firm age | *Age* | The year of company establishment |

### 3.3. Models

This investigation uses the ordinary least squares (OLS) regression model below to check the raised hypotheses. Model (1) was constructed to examine H1.

$$EP_{it} = \alpha_1 + \beta_1 RD_{it} + \beta_2 Control_{it} + \varepsilon_{1it} \tag{1}$$

To test H2 and H3, regression Models (2) and (3), with interaction terms, were constructed, respectively.

$$EP_{it} = \alpha_2 + \beta_1 RD_{it} + \beta_2 SAA_{it} + \beta_3 RD_{it} \times SAA_{it} + \beta_4 Control_{it} + \varepsilon_{2it} \tag{2}$$

$$EP_{it} = \alpha_3 + \beta_1 RD_{it} + \beta_2 Govsubsidy_{it} + \beta_3 RD_{it} \times Govsubsidy_{it} + \beta_4 Control_{it} + \varepsilon_{3it} \tag{3}$$

To examine the co-moderation impact of financing constraints and government subsidies and, specifically, to examine whether the moderating influence of financing constraints depends on government subsidies, Models (4) and (5) were constructed by using the model structures from Li et al. [50].

$$EP_{it} = \alpha_4 + \beta_1 RD_{it} + \beta_2 SAA_{it} + \beta_3 Govsubsidy_{it} + \beta_4 RD_{it} \times SAA_{it} + \beta_5 RD_{it} \times Govsubsidy_{it} + \beta_6 Control_{it} + \varepsilon_{4it} \tag{4}$$

$$EP_{it} = \alpha_5 + \beta_1 RD_{it} + \beta_2 SAA_{it} + \beta_3 Govsubsidy_{it} + \beta_4 RD_{it} \times SAA_{it} + \beta_5 RD_{it} \times Govsubsidy_{it}$$
$$+\beta_6 RD_{it} \times SAA_{it} \times Govsubsidy_{it} + \beta_7 Control_{it} + \varepsilon_{5it} \tag{5}$$

where $i$ indicates the enterprise, $t$ stands for the year, $EP_{it}$ is the environmental performance in that year, $RD_{it}$ is the corporate innovation in that year, $SAA_{it}$ is the financing constraints in that year, and $Govsubsidy_{it}$ is the government subsidies in that year. $RD_{it} \times SAA_{it}$ represents the interaction between corporate innovation and financing constraints, $RD_{it} \times Govsubsidy_{it}$ represents the interaction between corporate innovation and government subsidies, and $RD_{it} \times SAA_{it} \times Govsubsidy_{it}$ represents the three-way interaction of corporate innovation, financing constraints, and government subsidies. $Control_{it}$ was for the control variables, $\alpha_1 - \alpha_5$ for the intercept, $\beta_1 - \beta_7$ for the regression coefficients, and $\varepsilon_{1it} - \varepsilon_{5it}$ for the error terms.

## 4. Results

### 4.1. Descriptive Statistics and Correlation Analysis

The descriptive statistics and correlation analysis of the variables are displayed in Table 3. It is observed that the difference between the maximum and minimum values of the environmental performance is 9, with a standard deviation of 0.992, meaning that green development is not progressing well in the sample companies. The maximum value of corporate innovation is 12.457, significantly exceeding the average value, suggesting that the level of innovation varies greatly; so, it is urgent for companies to raise their innovation level. The mean value of financing constraints is 3.837, which is a small difference from the maximum value, indicating widespread difficulties faced by enterprises in financing. Government subsidies ranged between 0.000 and 0.246, revealing the unevenness of government subsidies to the sample firms.

**Table 3.** Descriptive statistics and correlations.

| Var | 1. EP | 2. RD | 3. SAA | 4. Govsubsidy | 5. ROA | 6. Cent | 7. Growth | 8. Lev | 9. Size | 10. Age |
|---|---|---|---|---|---|---|---|---|---|---|
| 1 | 1 | | | | | | | | | |
| 2 | 0.400 *** | 1 | | | | | | | | |
| 3 | −0.168 *** | 0.035 | 1 | | | | | | | |
| 4 | 0.339 *** | 0.109 *** | 0.043 | 1 | | | | | | |
| 5 | 0.053 * | 0.163 *** | 0.001 | −0.082 *** | 1 | | | | | |
| 6 | 0.198 *** | −0.053 * | −0.285 *** | −0.080 *** | 0.001 | 1 | | | | |
| 7 | 0.025 | −0.019 | 0.032 | −0.037 | 0.098 *** | 0.026 | 1 | | | |
| 8 | 0.531 *** | −0.202 *** | −0.039 | −0.146 *** | −0.358 *** | 0.145 *** | 0.021 | 1 | | |
| 9 | 0.544 *** | −0.254 *** | −0.262 *** | −0.191 *** | −0.058 * | 0.219 *** | 0.013 | 0.567 *** | 1 | |
| 10 | 0.135 *** | −0.039 | 0.586 *** | −0.003 | −0.084 *** | −0.085 *** | −0.006 | 0.148 *** | 0.110 *** | 1 |
| Mean | 5.437 | 1.820 | 3.870 | 0.008 | 0.040 | 35.564 | 0.363 | 0.451 | 22.678 | 18.239 |
| SD | 0.992 | 1.330 | 0.227 | 0.015 | 0.063 | 13.920 | 3.504 | 0.191 | 1.347 | 5.018 |
| Min | 2.000 | 0.000 | 2.318 | 0.000 | −0.531 | 8.087 | −0.643 | 0.035 | 19.199 | 3.000 |
| Max | 11.000 | 12.457 | 4.826 | 0.246 | 0.399 | 82.505 | 96.024 | 1.352 | 28.636 | 41.000 |

\* $p < 0.1$; \*\*\* $p < 0.01$ (two-tailed). $n = 1127$.

The correlation coefficient values range between 0.001 and 0.586 and variance inflation factor (VIF) values between 1.02 and 1.96, implying no multicollinearity problems among the variables.

### 4.2. Hypothesis Testing

The three research hypotheses of this study were examined by multiple regression models. The full-sample empirical results are shown in Table 4. The F statistics of all the models are significant. Models (1) to (5) indicate that the influence coefficients of all six control variables are positive. Firm size, leverage, and profitability were significantly and positively correlated with environmental performance. The regression coefficient of corporate innovation is significantly positive at the 1% level; so, H1 is verified. Hence, stimulating corporate innovation can help improve environmental performance.

**Table 4.** Full-sample regression results.

| Variable | EP | | | | |
|---|---|---|---|---|---|
| | (1) | (2) | (3) | (4) | (5) |
| RD | 0.131 *** | 0.143 *** | 0.137 *** | 0.143 *** | 0.147 *** |
| | (0.0113) | (0.0114) | (0.0108) | (0.0108) | (0.0110) |
| SAA | | 0.130 | | 0.111 | 0.131 |
| | | (0.0883) | | (0.0827) | (0.0833) |
| Govsubsidy | | | 0.134 *** | 0.133 *** | 0.132 *** |
| | | | (0.0105) | (0.0104) | (0.0104) |
| RD × SAA | | −0.258 *** | | −0.211 *** | −0.218 *** |
| | | (0.0538) | | (0.0531) | (0.0532) |
| RD × Govsubsidy | | | 0.02498 ** | 0.02726 *** | 0.0333 ** |
| | | | (0.00207) | (0.00215) | (0.00909) |
| RD × SAA × Govsubsidy | | | | | −0.0553 * |
| | | | | | (0.0305) |
| Roa | 2.014 *** | 1.830 *** | 1.602 *** | 1.468 *** | 1.478 *** |
| | (0.254) | (0.254) | (0.238) | (0.239) | (0.238) |
| Cent | 0.000985 | 0.00164 | 0.000606 | 0.00121 | 0.00114 |
| | (0.00107) | (0.00108) | (0.000992) | (0.00101) | (0.00101) |
| Growth | 0.000841 | −0.000270 | −0.000237 | −0.00121 | −0.00126 |
| | (0.00413) | (0.00409) | (0.00384) | (0.00382) | (0.00382) |
| Lev | 0.765 *** | 0.704 *** | 0.645 *** | 0.595 *** | 0.608 *** |
| | (0.1000) | (0.0999) | (0.0934) | (0.0937) | (0.0939) |
| Size | 0.592 *** | 0.620 *** | 0.575 *** | 0.598 *** | 0.598 *** |
| | (0.0136) | (0.0149) | (0.0128) | (0.0141) | (0.0140) |
| Age | 0.00845 | 0.00423 | 0.00902 | 0.00543 | 0.00491 |
| | (0.00292) | (0.00374) | (0.00271) | (0.00349) | (0.00350) |
| Constant | −5.603 *** | −6.153 *** | −5.145 *** | −5.592 *** | −5.584 *** |
| | (0.283) | (0.302) | (0.266) | (0.285) | (0.285) |
| N | 1127 | 1127 | 1127 | 1127 | 1127 |
| Adj-$R^2$ | 0.7637 | 0.7690 | 0.7956 | 0.7988 | 0.8014 |
| F | 37.449 | 40.740 | 41.738 | 48.811 | 52.089 |

* $p < 0.1$; ** $p < 0.05$; *** $p < 0.01$ (two-tailed).

Model (2) and Model (3) are used to test the moderating influences of financing constraints and government subsidies, respectively. The coefficient and significance level of the interaction term between the financing constraints and corporate innovation reveals that the financing constraints have a negative moderating influence on the relationship between corporate innovation and environmental performance. Thus, H2 passes the test. Meanwhile, the coefficient of the interaction term between the government subsidies and corporate innovation is significantly positive ($\beta > 0$, $p < 0.01$), showing that the government subsidies have a positive moderating effect on the relationship between corporate innovation and environmental performance. Thus, H3 is supported.

Model (4) illustrates that the effect of corporate innovation on environmental performance is moderated by both financing constraints and political government subsidies, and the results of the direction and degree of moderation are consistent with Models (2) and (3). The effects of corporate innovation on environmental performance are influenced by the firm's external capital environment. Well-funded companies, having reduced innovation risks, focus more on the development of innovation abilities. As such, these companies continue to improve resource allocation efficiency and environmental sustainability, resulting in a high environmental performance.

Model (5) adds the three-way interaction term of corporate innovation, financing constraints, and government subsidies to Model (4) to reflect the relationships between the moderating effects. The results suggested that the coefficient of the three-way interaction term between firm innovation, financing constraints, and government subsidies was −0.0553 and significant at the 1% level, implying that the influence of financing constraints on the relationship between corporate innovation and environmental performance varies

at different levels of government subsidies. This is mainly due to the fact that, on the one hand, the amount of turnover capital available for corporate innovation increases in the presence of government subsidies. On the other hand, government subsidies can convey positive information about the enterprise to the outside world and improve its reputation. Therefore, government subsidies are conducive to guiding external capital inflow, reducing the investment cost and financing pressure of corporate innovation, and prompting enterprises to take relatively more environmental responsibilities.

To demonstrate the moderating effects of the moderating variables (financing constraints and government subsidies), this study groups each moderating variable using the mean as the dividing line, as follows: the samples above one standard deviation of the mean are the high-level group of the moderating variable, while the samples below one standard deviation of the mean are the low-level group. The moderating effects after grouping are shown in Tables 1–4.

Figure 2 provides clearer evidence of the moderating effect of financing constraints. Corporate innovation is positively correlated with environmental performance for both the high and the low financing constraints groups. The slope of the low financing constraints group is greater than that of the high financing constraints group, indicating a more significant positive effect of corporate innovation on environmental performance with lower financing constraints.

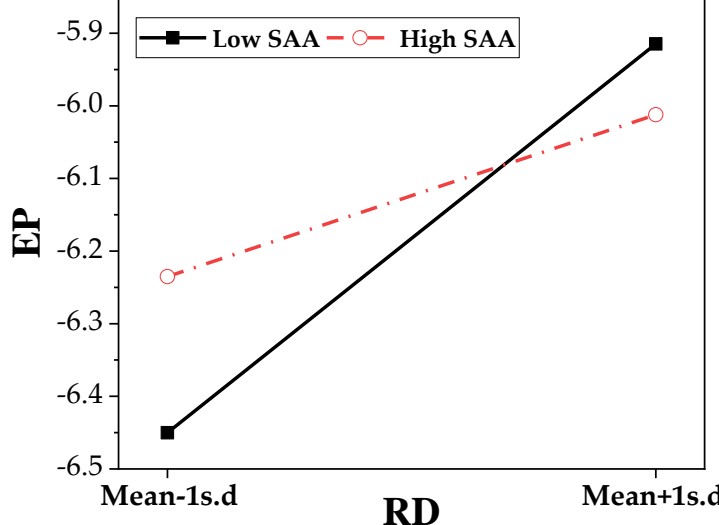

**Figure 2.** Moderation of the relationship between corporate innovation and environmental performance by financing constraints.

Figure 3 explicitly demonstrates the effectiveness of government subsidy moderation. The increase in environmental performance was higher in the high government subsidies group (environmental performance increased from −5.178 to −4.717) than in the low government subsidies group (environmental performance increased from −5.475 to −5.210), indicating that the higher the government subsidies of the firm are, the greater the positive effect of corporate innovation on environmental performance.

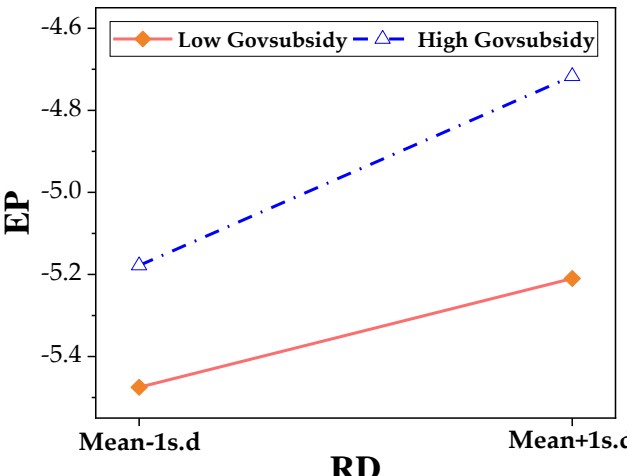

**Figure 3.** Moderation of the relationship between corporate innovation and environmental performance by government subsidies.

Figure 4 exhibits the co-moderation of financing constraints and government subsidies. The results indicate that the slope is greatest for the low financing constraints and high government subsidies groups, i.e., the effect of corporate innovation on environmental performance becomes more pronounced when the financing constraints are decreased and the government subsidies level is increased. Further comparing the two groups with high government subsidies (black solid line and blue dotted line), it can be concluded that the difference in the modulating effect of financing constraints narrows with a high level of government subsidies.

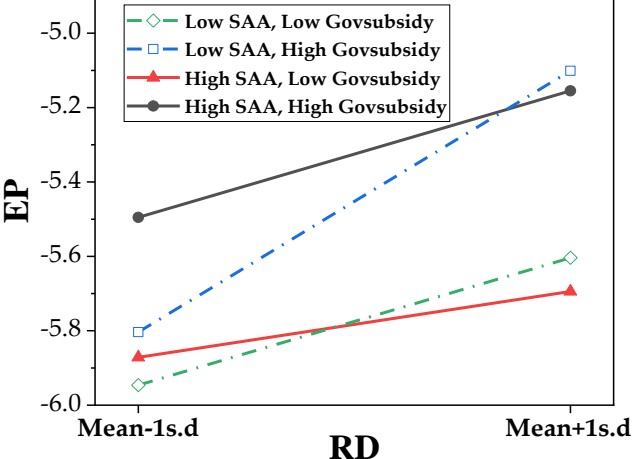

**Figure 4.** Co-moderation of the relationship between corporate innovation and environmental performance by financing constraints and government subsidies.

### 4.3. Robustness Tests

Selecting different measures of the critical variables may significantly influence the robustness of the conclusions. Patents are a widely used indicator of firm innovation [2,10]. Therefore, in the robustness test, corporate innovation was re-expressed as the number of corporate patents granted (denoted as *RD*2), as published by the China Research Data Service Platform (CNRDS database). To eliminate the industry-scale influences, the patents granted numbers by adding 1 were logarithmized. Table 5 shows the regression results after the substitution of corporate innovation with the new expression, showing that the regression coefficients have the same significance and direction (positive and negative) as in Table 4. Consequently, it was concluded that the findings of this study are robust.

**Table 5.** Robustness test.

| Variable | EP | | | | |
|---|---|---|---|---|---|
| | (1) | (2) | (3) | (4) | (5) |
| RD2 | 0.139 *** | 1.352 *** | 0.117 *** | 1.142 *** | 1.058 *** |
| | (0.0135) | (0.260) | (0.0138) | (0.244) | (0.249) |
| SAA | | 1.660 *** | | 1.410 *** | 1.354 *** |
| | | (0.329) | | (0.309) | (0.310) |
| Govsubsidy | | | 28.79 *** | 28.44 *** | 31.58 *** |
| | | | (4.124) | (4.089) | (4.506) |
| RD2 × SAA | | −0.311 *** | | −0.262 *** | −0.242 *** |
| | | (0.0667) | | (0.0625) | (0.0637) |
| RD2 × Govsubsidy | | | 0.0319 *** | 0.0318 *** | 0.0861 ** |
| | | | (0.00738) | (0.00732) | (0.0337) |
| RD2 × SAA × Govsubsidy | | | | | −0.0124 * |
| | | | | | (0.00751) |
| Roa | 2.133 *** | 1.983 *** | 1.750 *** | 1.630 *** | 1.619 *** |
| | (0.256) | (0.255) | (0.241) | (0.241) | (0.241) |
| Cent | 0.00125 | 0.00182 * | 0.000940 | 0.00146 | 0.00148 |
| | (0.00108) | (0.00110) | (0.00101) | (0.00103) | (0.00103) |
| Growth | 0.00287 | 0.000831 | 0.00119 | −0.000525 | −0.000576 |
| | (0.00419) | (0.00416) | (0.00392) | (0.00390) | (0.00390) |
| Lev | 0.827 *** | 0.768 *** | 0.724 *** | 0.676 *** | 0.679 *** |
| | (0.101) | (0.101) | (0.0951) | (0.0952) | (0.0952) |
| Size | 0.582 *** | 0.608 *** | 0.565 *** | 0.587 *** | 0.588 *** |
| | (0.0137) | (0.0149) | (0.0129) | (0.0141) | (0.0141) |
| Age | 0.00984 | 0.00566 | 0.0102 | 0.00649 | 0.00673 |
| | (0.00295) | (0.00378) | (0.00276) | (0.00354) | (0.00354) |
| Constant | −6.115 *** | −13.09 *** | −5.439 *** | −11.38 *** | −11.17 *** |
| | (0.303) | (1.408) | (0.289) | (1.325) | (1.331) |
| N | 1127 | 1127 | 1127 | 1127 | 1127 |
| Adj-$R^2$ | 0.7583 | 0.7633 | 0.789 | 0.7926 | 0.7929 |
| F | 36.022 | 39.21 | 40.451 | 46.872 | 50.556 |

* $p < 0.1$; ** $p < 0.05$; *** $p < 0.01$ (two-tailed).

## 5. Discussion

First, corporate innovation positively impacts environmental performance, agreeing with the findings of Abbas and Sagsan [27], Long et al. [26], and Wang and Liu [51], in contrast to Li and Lin [48] and Chen and Lee [10], who argue that corporate innovation may be ineffective in increasing environmental performance. From the viewpoint of Färe et al. [9], the growth of firm size caused by innovation still requires more resources to be invested and negatively impacts the environment. However, Churchill et al. [52] argue that the long-run development gains of corporate innovation may offset and compensate for the economic losses. The above disagreement is due to the fact that the economic structure is restructured and optimized by innovation in the long run, which contributes to the overall environmental benefits in society. In addition, corporate innovation can optimize the efficiency of resource allocation, thus achieving synergy between economic development and environmental management [53].

Second, financing constraints negatively moderate the effect of corporate innovation on environmental performance. While financing constraints would force companies to be careful with their capital and avoid unnecessary costs, they would also curtail eco-investment with high risks [54], making it difficult for companies to cope with increasingly stringent environmental regulations. If a company faces financing constraints, it will need to focus on exploring product markets and increasing business revenues. To avoid the higher financial risks associated with corporate innovation and to achieve the expected economic benefits, corporate investors may have to abandon or postpone innovative activities in environmental governance, and thereby financing constraints would dampen the impact of corporate innovation on environmental performance. According to Noailly and Smeets [55],

environmental innovation projects require adequate and stable financial support, by which the decision to initiate environmental innovation projects may be hindered when firms face financing constraints. Faced with financing constraints such as few financing sources and a high cost of employed capital, the capital required for corporate innovation may exceed the firm's affordability; so, to reduce the marginal cost and uncertainty of corporate innovation, firms would forego the long-term benefits of environmental innovation in noncore businesses [56]. Facing stronger environmental regulation, enterprises with low financing constraints have access to stable innovation funding, thus making them more likely to implement environmental policies and balance environmental protection and development [57].

Third, government subsidies positively regulate the impact of corporate innovation on environmental performance. Government subsidies play a vital role in promoting corporate innovation and improving corporate compliance with environmental responsibility. However, agency theory suggests that enterprises may not allocate government subsidy resources efficiently to control environmental pollution. Ineffective enforcement of government laws and regulations and inadequate pre-surveys in issuing government subsidies can also lead to less efficient allocation of government subsidy resources [19]. However, this study finds that government support for corporate innovation enhances a firm's sense of environmental responsibility; so, firms are more likely to perceive a clear return on environmental investment, thus shifting their innovation strategies to environmental projects for long-term environmental benefits. In addition, the higher the level of government subsidies an enterprise receives, the higher the level of technological innovation recognition for its research projects, which gives it more confidence and ability to fulfill the government's environmental requirements.

Fourth, the impact of financing constraints on the relationship between corporate innovation and environmental performance varies at different levels of government subsidies. Increased levels of government subsidies mitigate the weakening effect of financing constraints on the focal relationship. When the level of government subsidies is low, financing constraints are not effectively mitigated, with the result that firms may be reluctant to undertake environmental innovation activities entirely by themselves, making it difficult to achieve high environmental performance. This is because intense market competition forces enterprises to focus primarily on profit maximization or cost minimization [25] rather than the environmental management of noncore businesses. However, firms with urgent capital needs but high financing constraints have more innovation willingness and ability to take advantage of the firms' resources efficiently, toward high environmental performance as government subsidies increase. Thus, for corporations with high government subsidies, the inhibitory effect of financing constraints on the focal relationship is weakened.

## 6. Conclusions and Implications

### 6.1. Conclusions

Using 1127 observations of energy-intensive Chinese listed companies from 2012 to 2020, this research analyzes the influence of corporate innovation on environmental performance in the light of reputation theory, resource base theory, sustainability theory, and reputation theory and investigates the impact of financing constraints and government subsidies on the relationship between corporate innovation and environmental performance from the viewpoint of the enterprise's external capital environment, as well as the variations of the moderating effects of financing constraints with different levels of government subsidies. The following conclusions are drawn: first, corporate innovation positively contributes to environmental performance. Second, the regulating impact of financing constraints is significantly negative, while government subsidies positively regulate the relationship between corporate innovation and environmental performance. Finally, the degree of the negative moderation of the financial constraints varies across levels of government subsidies, with high levels of government subsidies weakening the

negative impact of financial constraints on the relationship between corporate innovation and environmental performance.

### 6.2. Implications

First, the empirical results demonstrate that corporate innovation is a strong driver of environmental performance. The recent commitment of China to achieve carbon neutrality by 2060 requires extensive investment in environmental programs and technologies [58]. Therefore, enterprises should strengthen the transformation, application, and dissemination of technological achievements to realize a dual-win situation of economic growth and environmental pollutant reduction. Notably, innovation activities are time-consuming and require long-term financial support. Therefore, the impact of financing constraints and government subsidies should not be underestimated.

Second, financing constraints can weaken the focal relationship in this study. Therefore, the Chinese government should establish a diversified financing platform to provide efficient services for innovation throughout the cycle to promote green production in enterprises. Financial institutions are required to increase financing support for corporate innovation and improve the financing environment to enhance the accuracy of the financial policies. In addition, financial institutions should improve the credit system for corporate innovation and provide comprehensive supervision and management of corporate financing practices to ensure the efficiency of the use of corporate financing funds.

Third, government subsidies facilitate the positive influence between corporate innovation and environmental performance; therefore, the allocation of government subsidies must be more effective and scientific. On the one hand, the Chinese government should establish and improve the standards and processes of subsidy application and review, improve the payment methods of subsidies, and ensure that the amount of subsidies received by enterprises is in accordance with their actual needs to raise the effectiveness of the government subsidy allocation. On the other hand, the Chinese government ought to enhance the construction of a subsidy supervision mechanism and perfect the market financial environment to avoid the rent-seeking behavior of enterprises, with the purpose of promoting the green development of enterprises.

Fourth, the results of the co-moderation suggest that raising government subsidies while mitigating financing constraints, i.e., combined government and financial institution support, can improve the firms' external capital environment, prompting them to rationally allocate innovation resources to maximize environmental performance. Enterprises should expand their financing sources to mitigate financing constraints and optimize government subsidy allocation efficiency in accordance with the government's environmental requirements. When providing subsidies, the Chinese government should not only consider the financing environment of enterprises but also favor enterprises with proactive environmental protection that are investing external capital more effectively in environmental performance activities.

### 6.3. Limitations and Research Prospects

Listed below are some limits of this research: first, corporate innovation is denoted in two ways, using innovation expenditures as well as the number of corporate patents granted. However, this study does not distinguish between innovation processes and innovation outcomes, and subsequent studies can further classify corporate innovation and test the robustness of our results. Second, this study explored whether corporate innovation influenced environmental performance in the current year. However, the environmental effects of corporate green development have a long time lag. The evolution of environmental performance and its consequences is a future direction for subsequent research.

**Author Contributions:** Conceptualization, H.D. and C.L.; methodology, H.D.; validation, H.D., C.L. and L.W.; formal analysis, L.W.; investigation, H.D.; resources, C.L.; data curation, C.L.; writing—original draft preparation, H.D.; writing—review and editing, C.L.; visualization, L.W.; supervision, L.W.; funding acquisition, C.L. and L.W. All authors have read and agreed to the published version of the manuscript.

**Funding:** The research is supported by National Social Science Fund of China, grant number 21BJY217; Innovation Strategy Research Plan Project of Fujian Province, grant number 2021R0069.

**Institutional Review Board Statement:** Not applicable.

**Informed Consent Statement:** Not applicable.

**Data Availability Statement:** Not applicable.

**Conflicts of Interest:** The authors declare no conflict of interest.

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
