# Peer review of "The Impact of Corporate Innovation on Environmental Performance: The Moderating Effect of Financing Constraints and Government Subsidies"

_sustainability, doi:10.3390/su141811530_

Round 1
Reviewer 1 Report
After review of the manuscript “The impact of corporate innovation on environmental performance: The moderating effect of financing constraints and government subsidies” I would like to congratulate you and your team for doing such a good research work in your submitted paper for publication in this prestigious journal. The topic is very interesting and I liked the topic and I personally like to appreciate your efforts to present your research work in such a nice manner. But before your work will be recommended or will be given any possible acceptance few comments must be incorporated to improve the quality of your work as well as for further publication in this reputed journal. I have the following major observations or queries and comments which may further enhance your piece of work. The authors require to modify the following points in detail.
- Some equations and tables are out of the pages; this issue should be fixed. (see page 7 and page 8)
- The introduction part is required adding a few more sentences to increase the strength of this article.
- Moreover, what about the novelty of the study? This must be underlined in the study as well as its contribution to the literature.
- The authors might use an abbreviation table because the authors used lots of abbreviations in the text. In its current form, it is difficult to read.
- To increase the readability of the study, before the conclusion and implications sections, the authors can construct a graphical illustration of the main finding.
Reviewer 2 Report
1. The introduction section needs to add a more updated background to the problem with a more real context, so that the needs of the research object can be described better.
2. It is necessary to add references that strengthen the urgency of this research, especially regarding the importance of subsidies and financial factors which are additional variables in the research model.
3. The construction of the hypothesis needs to be rearranged, so that the links between the constituent references become more coherent and clear, and need to be included in the conceptual model (Figure 1). This includes the variables used in the proposed or new research model.
4. The data collection process needs to be explained in more detail, due to the long duration and large amount of data, thus ensuring that the data is valid.
5. Conclusions need to be synchronized with the problem statements presented in the introduction and background
6. Need to re-check the writing in English, so that it is more proper
Reviewer 3 Report
The paper presents an interesting issue using ordinary least squares to investigate the relationship between the external capital environment, corporate innovation, and environmental performance of Chinese firms. The analysis is quite detailed and their results appear to be very consistent and robust across their specifications. I have read with interest the introduction and the goal of the paper seems clear and concrete. I believe, also, the authors have made a carefully constructed literature review.
Round 2
Reviewer 2 Report
All revisions made are in accordance with the comments and input submitted